# Single Incision Pediatric Endoscopic Surgery: From Myth to Reality a Case Series

**DOI:** 10.3390/medicina55090574

**Published:** 2019-09-07

**Authors:** Bradley J. Wallace, Raphael N. Vuille-dit-Bille, Ahmed I. Marwan

**Affiliations:** 1Division of Pediatric Surgery, Children’s Hospital Colorado, University of Colorado, Anschutz Medical Campus, 13123 East 16th Ave, B323, Aurora, CO 80045, USA (B.J.W.) (R.N.V.-d.-B.); 2Division of Pediatric Surgery, University Children’s Hospital of Basel, 4056 Basel, Switzerland; 3Division of Pediatric Surgery, University of Basel, 4056 Basel, Switzerland

**Keywords:** single-incision, pediatric surgery, laparoscopic, minimally invasive

## Abstract

Laparoscopic surgery has continued to evolve to minimize access sites and scars in both the adult and pediatric populations. In children, single-incision pediatric endoscopic surgery (SIPES) has been shown to be effective, feasible, and safe with comparative results to multiport equivalents. Thus, the use of SIPES continues over increasingly complex cases, however, conceptions of its efficacy continue to vary greatly. In the present case series and discussion, we review the history of SIPES techniques and its current application today. We present this in the setting of five common myths about SIPES techniques: limitations against complex cases, restrictions to specialized training, increased morbidity outcomes, increased operative lengths, and increased operative costs. Regarding the myth of SIPES being limited in application to simple cases, examples were highlighted throughout the literature in addition to the authors’ own experience with three complex cases including resection of a lymphatic malformation, splenectomy with cholecystectomy, and distal pancreatectomy with splenectomy. A review of SIPES learning curves shows equivalent operative outcomes to multiport learning curves and advancements towards practical workshops to increase trainee familiarity can help assuage these aptitudes. In assessing comorbidities, adult literature reveals a slight increase in incisional hernia rates, but this does not correlate with single-incision pediatric data. In experienced hands, operative SIPES times average approximate multiport laparoscopic equivalents. Finally, regarding expenses, SIPES represents an equivalent alternative to laparoscopic techniques.

## 1. Introduction

Single incision pediatric endoscopic surgery (SIPES) is a minimally invasive endosurgical technique that is increasingly being performed by pediatric surgeons globally, with continued development towards the variety of operations for which it can be adopted [1,2]. After a transition of proven safety and equivalent to improved outcomes over morbidity, mortality, length of stay, postoperative pain requirements, and cosmesis, amongst other results, the use of SIPES has grown rapidly [3].

Historically, since Georg Kelling’s 1901 approach of “Celioscopy” and Hans Jacobaeus’s 1910 first use of the laparoscopic approach in humans for diagnostic aid, the breadth over which laparoscopic procedures have been performed has continuously evolved towards increasingly complex procedures [4]. Initial forays into laparoscopic techniques soon arose including adhesiolysis and diagnostic biopsy, with the first single-incision laparoscopic bilateral tubal ligation performed in 1968 by Dr. Clifford Wheeless, and the first multiport laparoscopic organ resection, a salpingectomy, performed by Dr. Tarasconi in 1975 [4,5]. The eventual development of the computer chip allowed for projection of imaging to a screen allowing for the more modernized practice of laparoscopic surgery, subsequently prompting the first laparoscopic cholecystectomy in 1985 by Dr. Erich Mühe, and the first laparoscopic appendectomy in 1988 by Dr. Kurt Semm [4,6,7]. Transitioning to single-incision techniques beyond the simple ligations and biopsies, the single-port approach arose for appendectomies with Dr. Marco Pelosi as early as 1992, for cholecystectomies with Dr. Navarra in 1997, and the first pediatrics single-incision appendectomy by Dr. Begin and SIPES gastrostomy by Dr. Keith Georgeson both in 1993 [3,4,8,9]. Additionally, as prevalence of SIPES continued to increase, its growth in the literature has mainly focused on appendectomies, cholecystectomies, alimentary procedures such as gastrostomies and jejunostomies, and herniorrhaphies [2,4,10,11]. Recently, the idea of single-incision laparoscopic surgical access for more complex cases such as colectomies, hepatectomies, or pancreatectomies has arisen, though is less practiced [12,13,14,15].

While multiport laparoscopic access moved beyond initial resistant preconceptions against the approach, it has since evolved into the currently ubiquitous technique for many surgically treatable diseases, including several cancers [16,17,18,19,20]. In contrast, single-incision surgery techniques remain used by only a small subset of providers, regardless of current evidence of equivalent outcomes [19]. Despite hesitancy against, previous studies have shown the safety of single-incision techniques for several procedures [2,3,11]. Reported improvements of patient length of stay and non-inferior operative time and complications continue to stem controversy, with results such as operation time and complication rates shown to correlate to individual surgeon experience [3,6,10,21]. The present manuscript aims to review biases against single-incision surgery and discuss the evidence about them in the setting of three case reports exemplified below.

## 2. Case Report

### 2.1. Case 1: SIPES-Assisted Mesenteric Lymphatic Cyst Excision with Ileocecectomy and Primary Anastomosis

A Colorado Fetal Care Center consultation during a pregnancy otherwise complicated by hyperemesis showed a fetal abdominal cyst for which multidisciplinary counseling was performed. Differential diagnoses including ovarian cyst, choledochal cyst, intestinal duplication, and lymphovascular malformations were discussed. Postnatal ultrasound at age 3 weeks revealed a right lower quadrant cyst, which initially appeared to be an intestinal duplication cyst, for which subsequent intervention was planned. At 3 months of age, the child underwent surgery. A GelPOINT^®^ Mini Advanced Access Platform (Applied Medical Resources Corporation, Rancho Santa Margarita, CA, USA) was used with establishment of an 8 mm Hg pneumoperitoneum. An 8.5 cm multi-loculated lymphatic cystic lesion was identified in the mesentery integrated with the ileocolic junction. Due to its size a spinal needle was introduced under direct laparoscopic vision to obtain and send the lymphohemorrhagic fluid for cell count and cytology. Once determined to be a true mesenteric cyst, the suction catheter was used for further decompression, which allowed for anatomic delineation and delivery of the ileocecum via the port site. An extracorporeal ileocecectomy was performed with a hand-sewn anastomosis, the mesenteric defect was closed, and viscera returned. Fascia and skin were closed and by evening of the surgery he was tolerating a regular diet with normal bowel function. Estimated blood loss was less than 2 mL and total operative time was 2 h 53 min. He was discharged the following day without complications, and pathology was consistent with a lymphatic malformation.

### 2.2. Case 2: SIPES Splenectomy and Cholecystectomy

A 12-year-old female with hereditary spherocytosis with cholelithiasis and 16 cm palpable splenomegaly was referred for elective splenectomy and cholecystectomy. After preoperatively completing her vaccines, she was later brought to the operating room, placed supine, induced with general anesthesia, lifted with a small bump under the left flank, and secured with tape. A 1.5 cm vertical incision at the base of the umbilicus was made and the Olympus QuadPort+ (Olympus Corporation of the Americas, Center Valley, PA, USA) was placed prior to establishment of 14 mm Hg pneumoperitoneum. Reverse Trendelenburg position, with more left side up was used to first perform the splenectomy. The stomach was mobilized medially with takedown of the gastrosplenic ligament using an ENSEAL^®^ device (Ethicon US, L.L.C., Somerville, NJ, USA; Figure 1). The superior splenophrenic ligaments were taken down completely prior to takedown of the splenocolic and splenorenal ligaments. The hilum was then dissected, with identification of a single splenic artery and vein using Maryland forceps to create clear windows. An Endo-GIA stapler with 3.0 mm, 2.5 mm, and 2.0 mm staple lines was used to fire across the hilum with assurance of hemostasis. Splenic dissection was completed with the ENSEAL^®^ and attention was brought to the gallbladder.

At this point the patient was placed in reverse Trendelenburg, now with the right side up, and an additional 5 mm incision was made in the right upper quadrant for cephalo-lateral retraction of an extremely adherent intrahepatic gallbladder. Via the QuadPort+ the Hartmann’s pouch was retracted caudo-laterally and hook electrocautery was used to dissect off the peritoneum and dissect the triangle of Calot to obtain the critical view. The small cystic artery was taken down with hook electrocautery, and cystic duct laparoscopically clipped. The gallbladder was then dissected off the liver bed. The spleen was placed in an Endobag and morcellation accomplished inside the bag with a pair of ring-forceps prior to retrieval without spillage. The gallbladder was retrieved prior to port removal. Pneumoperitoneum was released, and fascia and skin closed. Estimated blood loss was less than 5 mL, and the operative time for the splenectomy was 2 h with an additional 67 min for the cholecystectomy. The patient was discharged on postoperative day two without complications and with resolution of symptoms.

### 2.3. Case 3: SIPES Distal Pancreatectomy with Splenectomy

A 14-year-old otherwise healthy female presented acute epigastric/left upper quadrant abdominal pain with normal labs. Computed tomography scanning revealed a 4 cm homogenous distal pancreatic mass of low Hounsfield units, abutting the splenic vein though seemingly without distinct vessel involvement, also without further masses, surrounding ascites, ductal dilation, nor lymphadenopathy. Upon surgical consultation, we recommended a distal pancreatectomy with possible splenectomy (vaccines were given prior to surgery). In the operating room she was positioned supine with an nasogastric tube placement. A 1.5 cm incision was made at the base of the umbilicus followed by placement of a Gelport and establishment of 15 mm Hg pneumoperitoneum. On complete inspection there was no evidence of metastases. The lesser sac was entered, with visualization of the pancreatic body and tail, including the mass. The splenic flexure was mobilized using an ENSEAL^®^ device along the greater stomach curvature towards the hilum, with subsequent electrocautery dissection of the inferior border of the pancreas taken laterally towards the spleen. Further posterior dissection was performed, however, after full elevation of the distal pancreas it was evident that the splenic vein was tightly adherent to the mass.

The decision was made to perform an additional splenectomy and two additional 5 mm step ports were added to the right and left upper quadrants to help in exposure. Short gastrics were then taken down via the ENSEAL^®^, with hilar dissection using the Maryland and vascular Endo GIA™ stapler load (Medtronic Minimally Invasive Therapies, Minneapolis, MN, USA). The distal pancreas was then dissected posteriorly to the inferior mesenteric vein, with good margins prior to stapling across with a 3.5 mm 60 mm load. There appeared to be hemostasis, and fibrin sealant was added to the staple line. The pancreatic specimen was placed in an Endobag, though it was unable to be retrieved via the umbilical incision. A 3 cm Pfannenstiel incision was then created to retrieve the pancreatic specimen en block, with subsequent closure of the incision after retrieval. Pneumoperitoneum was then reestablished, with completion of the splenic dissection and retrieval after morcellation within the bag, also without spillage. All port sites were closed, and the patient was brought to recovery after placement of an epidural. Estimated blood loss was 100 mL and total operative time was 6 h 7 min. She recovered without issues and was discharged on postoperative day four. Pathology revealed solid pseudopapillary neoplasm, with no further masses found on subsequent surveillance imaging.

## 3. Discussion

Mirroring the growth of practice within the adult literature equivalents, single-incision pediatric endoscopic surgery has been increasing in application over a greater variety of cases in children, including in: appendectomy, cholecystectomy, pyloromyotomy, splenectomy, alimentary procedures, gynecological procedures, herniorrhaphy, and pancreatectomy [2,4,22,23,24,25,26,27]. The three SIPES cases above include a cystic lymphatic malformation resection, splenectomy with cholecystectomy, and pancreatectomy with splenectomy. These highlight the use of SIPES over complex, though not infrequent, cases seen by pediatric surgery, and serve to lead a discussion of five common myths regarding SIPES practice.

### 3.1. Myth 1: SIPES Practice Is Only Applicable to Simple Cases

Single-incision surgery use in the pediatric realm has expanded beyond Dr. Georgeson’s SIPES gastrostomies described in 1993 and Dr. Esposito’s single trocar appendectomies described in 1998 to further include cholecystectomies, splenectomies, pyloromyotomies, intestinal surgeries, and gynecological procedures [1,28,29]. The use of SIPES for pancreatic surgery in children is more limited in the literature, however, case 3 correlates with the feasibility shown in the rare case reports on the technique [30,31]. The cases above show more complex cases that surgeons dedicated to the practice of SIPES can perform after they are more practiced and expert with the technique.

As is the case with “standard” multiport equivalents, during single-incision approaches to difficult cases, the additional advantage of one or more assist ports for exposure is always available [32]. Notably, the benefits of having a single-incision approach, should not preclude the addition of ports or access for forceps for retraction or exposure, as seen in cases 2 and 3. In these instances, an extra site allowed for improved retraction of the gallbladder cephalad and laterally, as well as allowed for safer dissection of the distal pancreatic mass with splenic attachment. This highlights how some patients’ anatomies combined with the technicality of having a single focal point for the instrument port may restrict some movements to a degree that may benefit from an assisting port [32,33,34]. This additional option does not require SIPES approach to be aborted in full and exists with any equivalent multiport technique as well. Its prudent use may save any laparoscopic case from having to open.

Other adaptive techniques that can be used in a SIPES approach include diagnostic evaluation, cystic decompression, and extracorporeal bowel manipulation, as seen in case 1, as well as endoscopic dissection, resection, morcellation, and ligation, as seen in cases 2 and 3. While these highlighted cases are more complex in nature, as surgeons become increasingly familiar with the technique of a single-incision approach, their practice can grow to encapsulate many of the cases they would otherwise approach multiport laparoscopically. Furthermore, case 2 highlights how a centralized single access approach from the umbilicus can allow for multiple operations over opposing abdominal quadrants which can be performed utilizing the same anesthetic and port site. SIPES can be adapted for a wide range of cases over a breadth of operative fields.

### 3.2. Myth 2: SIPES Practice Requires an Inordinate Amount of Training, with an Excessive Learning Curve

Incorporation of a SIPES approach into practice has been somewhat limited by perceptions of technical difficulty of the new approach when compared to a more “traditional” multiport approach. Whereas the port sites in a multiport technique are typically spaced apart from each other and placed in a triangulated fashion towards the target in order to optimize ergonomics, precision, and effectiveness of motion, in a single-incision approach the single incision forces ports to remain close [20]. Instead, this single-port access creates a single-port fulcrum point, beyond which distal triangulation must be performed [35]. Furthermore, the crossed instruments traveling through the single-port fulcrum result in the left hand movements affecting the screen-right sided instrument and conversely the right hand affecting the screen-left one. These movements of the lateral hand causing opposite-sided motion on screen creates an effect that can cause even the most practiced multiport laparoscopic expert surgeon to initially feel out of their normal routine. However, studies show that expert multiport laparoscopic surgeons do have an easier time quickly adapting to these differences of the single-incision approach [36]. Furthermore, in SIPES, articulation of instruments along the shaft, head, and handle (including flipping the handles upside down), as well as technological improvements towards smaller instrument diameters and port sizes, have allowed for both more mobility in exposure and instrument movement, thus minimizing restrictions from the fulcrum [35].

A learning curve does exist for practice of SIPES prior to obtaining full operative familiarity, just as a learning curve exists in multiport approaches [1,8,36,37,38]. Assessment of single-port learning curves suggest anywhere between 10 and 40 cholecystectomy cases are required prior to significant operative length improvements, and near 100 cases for appendectomy time improvements [39,40]. Interestingly, experience in conventional laparoscopy may overlap into improved familiarity with the single-port technique. Some studies in adult literature suggest that expert operators in multiport laparoscopic may begin their single-incision learning curve at an already near-proficient skill level [40]. Notably, single-incision comparisons of what constitutes a significant difference between expert and novice operative times and skilled task times may range, some reporting as low as a 5 min operative difference of performing an appendectomy, or as high as a 4 min task difference for every cutting or suturing task performed by the expert compared to the novice [37,41]. While mathematically “significant” these differences may not hold as much clinical significance, depending on the setting. Both multi- and single-incision studies have shown some correlation between faster learning curves and previous video game skill levels as well [38,42].

A component of the rapid development of multiport laparoscopic surgery has been workshops for trainees, such as the Fundamentals of Laparoscopic Surgery course offered by Society of American Gastrointestinal and Endoscopic Surgeons (SAGES) and the American College of Surgeons (ACS) [35]. Similarly, computer-based workshops also exist worldwide [43]. Adult single-incision workshops do exist, though are not as ubiquitous in training programs, which may perpetuate trepidation against its use [41]. In these workshops, participants can develop their efficiency and hands-on practice to more quickly adapt to the single-port specific skills used. Workshops for SIPES have only recently developed but have been used to show skill improvement of fellows, residents, and medical students for SIPES [38]. Possible apprehension regarding technique difficulty should not be a contraindication against use of SIPES, as trainees of all surgical levels have shown capability for learning.

### 3.3. Myth 3: SIPES Practice Causes Increased Complications

Initial comparative studies between single-incision laparoscopic surgery and multiport laparoscopic equivalents over adult patients in several institutions revealed increased conversions to open, operative times, and port-side hernias [44,45,46]. However, results from more recent randomized controlled trials are conflicting, with the only Cochrane review stating that no definitive conclusions can be drawn at this time [47,48]. Less evidence exists in pediatric single incision versus multiple incision laparoscopic approaches, however, the overwhelming results show comparatively equivalent conversion rates to open, hernia rates, and overall postoperative morbidities [40,49,50,51,52]. Naturally, these outcome studies may be limited in part by publication bias, however, the results stand in contrast to earlier publications in adults.

There is a growing body of evidence regarding the advantages SIPES has over multiport laparoscopy. While initial evaluations in adult literature seemed to show increased pain with a single-incision approach, this may be outdated by the larger incisions and equipment required by the approach at its nascence [44,53,54]. Studies in children have shown equivalent or even decreased pain scores with SIPES [53,55]. The improvements that have allowed for smaller trocars and instruments through a consequently smaller fascial incision have been postulated to cause decreased pain secondary to decreased abdominal wall trauma, in addition to the improved cosmetic outcome they provide [2,53,55,56]. Furthermore, typical pediatric incisions are naturally smaller than adult equivalents. Increasingly, larger comparisons over patients with single versus multiport approaches over appendectomies have shown perioperative outcomes of pain are noninferior to decreased in single site approaches, though the degree of significance in some studies is as small as a single analgesic dose, which again may not be as significant clinically [57,58,59]. Despite possible perceptions to the contrary, the possible superiority offered by SIPES especially regarding pain and cosmesis, in the setting of otherwise proven equivalence of morbidity rates in the pediatric literature, may be reason for favoring a SIPES approach compared to a multiport equivalent.

### 3.4. Myth 4: SIPES Practice Takes Longer Time in the Operating Theatre

Literature evaluating operative times of single-port against multiport approaches vary by operation, though overall most are comparable in timing. Meta-analyses for adult operations performed by experienced operators show comparative operative times in colectomies, nephrectomies, and splenectomies [35,47,60]. The largest comparative studies of adult appendectomies and cholecystectomies over these techniques seem to weigh towards increased operative time using a single-incision approach (cholecystectomies most commonly ranging 10 min longer, and appendectomies ranging 6 min longer); however, there is significant statistical heterogeneity in these comparisons [8,61,62]. Naturally, all of these studies can be subject to self-selection bias.

However, comparative studies published for single-incision operative times in children largely report comparative operative times, though this is limited by a paucity of high-level evidence. One study, including non-expert colectomy times for single-incision colectomy, showed 30 min longer operative time, however, when only experienced single-incision surgeon times were compared against multiport times there was no difference [63]. The largest single institutional SIPES review reported an operative time increase for SIPES appendectomy of 7 min, a 9 min difference for cholecystectomy, 4 min for pyloromyotomy, and an 8 min difference for splenectomy [2]. Heterogeneity of meta-analysis results for SIPES appendectomy times range from no difference to a 7 min difference overall [3]. These outcomes both reinforce the aforementioned learning curve as well as correlate with multiport laparoscopic operative times [64,65]. SIPES operative times can be variable, as can multiport times, and similarly greater SIPES expertise in technique allows for continued improvements with faster and safer operative times [2,10,40,41,66].

### 3.5. Myth 5: SIPES Practice Is More Expensive than Multiport Equivalents

Comparative financial studies have shown SIPES to be equivalent in cost to multiport approaches for common pediatric surgical diseases [2,44,67]. Assessments of both costs from operative time, as well as costs of operative tools, show equivalency over these disease states, with other studies showing equivalent hospital lengths of stay, returns, and complications [2,44,57,67,68]. Initial increased costs for single-incision approaches reported in literature have been minimized with the advent of standard reusable laparoscopic instruments [67,68].

Alternative methods shown to decrease cost include using conventional multiport equipment to a SIPES approach (i.e., using multiport trocars and instruments, which have shown to be adaptable in an effective and safe manner [69]). Notably, studies showing equivalency of SIPES versus multiport costs come from institutions with comparable operative times to multiport operative times, thus outlier institutions with longer SIPES times may have increased total procedural costs with the technique [68]. Overall, evaluation of cost between SIPES equipment against multiport equipment has comparable results, with dependent costs of operative times showing similar calculated comparisons [44,57,67,68].

## 4. Conclusions

Once the activation energy of overcoming the learning curve has been accomplished, a growing number of cases have shown SIPES can be safely applied over a variety of cases, with noninferior objective outcomes of incisional hernia rates, conversions to an open approach, comparative morbidities and recovery times, and possible superiority of pain and cosmesis [57]. Given the literature supporting SIPES can be subject to publication bias, examples of the extent of evidence regarding its use and outcomes have been highlighted over the various myths and preconceptions above. Nevertheless, the present review reflects on the interpretation of the limited literature about single-incision surgery in pediatric patients and is hence prone to bias itself. Limited evidence for SIPES approaches regarding feasibility, safety, and comparative results to multiport equivalents continues to grow with increased surgical application over increasingly more complex cases. Despite myths and perceptions that may arise when compared to multiport equivalents, SIPES can be performed over a comparable range of cases, with straightforward training, equivalent complication rates, similar case lengths, and analogous cost.

## Figures and Tables

**Figure 1 medicina-55-00574-f001:**
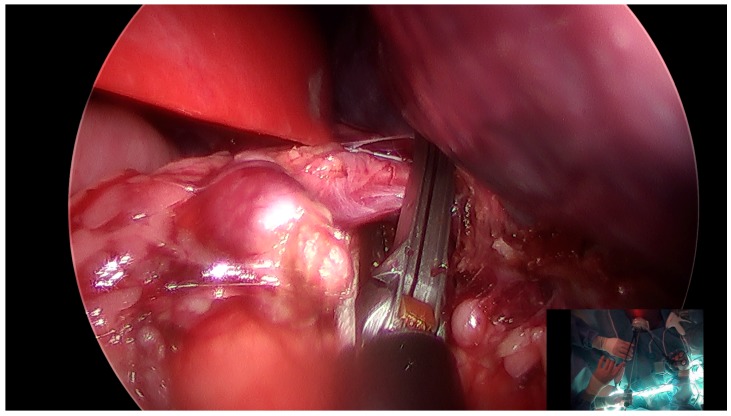
Takedown of the gastrosplenic ligament and short gastric vessels for splenic dissection.

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
