# Peer review of "Single Incision Pediatric Endoscopic Surgery: From Myth to Reality a Case Series"

_medicina, 2019, doi:10.3390/medicina55090574_

Round 1

Reviewer 1 Report

The manuscript medicina-569284 deals with the topic of single incision pediatric endoscopic surgery (SIPES). Their result suggested non-inferiority or acceptable outcomes of SIPES compared with multiport surgery, especially in the point of learning curve, complications, application for complex cases, operation time and costs. While their suggestions are interesting, there are several points that need further clarification.

Major points                                                  

This study suggested non-inferiority of SIPES, however superior points of SIPES were not described clearly. What is the reason for recommending SIPES instead of multiport surgery? This is an essential agenda for readers. The authors showed three complex cases who underwent SIPES. However, it required great deal of skill in order to perform SIPES for complex cases. It seems difficult to perform such an operation in all institutions. Authors should mention about it.  

Minor points 

   In three cases, operation times and amount of blood loss should be described.

Author Response

We appreciate the time and efforts by the editor and reviewers made towards this manuscript. We have addressed all issues indicated in the review report, as outlined below, and believe the revised version meets the journal publication standards.

Response to Comments from Reviewer 1

Major points:

This study suggested non-inferiority of SIPES, however superior points of SIPES were not described clearly. What is the reason for recommending SIPES instead of multiport surgery? This is an essential agenda for readers. The authors showed three complex cases who underwent SIPES. However, it required a great deal of skill in order to perform SIPES for complex cases. It seems difficult to perform such an operation in all institutions. Authors should mention about it.

Minor points:

In three cases, operation times and amount of blood loss should be described.

Responses:

Thank you for your efforts in careful review of our report along with the valuable input.

Indications for how SIPES can be superior to multiport equivalents have been added to the second paragraph of Myth 2 (lines 134-137), specifically regarding decreased pain and improved cosmesis. These superiorities were again added to the concluding statements paragraph at the end of the manuscript (line 298). We agree that the cases highlighted in Myth 3 do require a complexity of skill beyond the more common uses for SIPES such as appendectomies and gastrostomies. We have now acknowledged the needed expertise in their use (lines 229-231, lines 244-245). As suggested by the reviewer, we have added operative times and estimated blood losses to each of the cases listed (lines 166-167, lines 191-192, lines 218-219).

Reviewer 2 Report

The manuscript deals with an interesting argument, the single incision pediatric endoscopic surgery (SIPES) and its applications today, aiming to review biasis against single incision surgery and discuss the evidence about them.
In particular, the authors discuss five common myths about SIPES techniques: restrictions to specialized training, increased morbidity outcomes, limitations against complex cases, increased operative lengths, and increased operative costs.
This kind of structure could be innovative and catchy, however, reading the manuscript, some major concerns have emerged:
- It is not clear at all if the authors intended to write a review of literature, a case report or an article presenting their case series, resulting in a totally confusing work. Indeed, the authors presented the manuscript as a review of literature but, for example in rows 112-115 they write “In our own experience at a quarternary institution with surgical trainees involved in over 270 cases, we have not shown any increases in conversions to open (1.10%), SSI (3.3%) or otherwise morbidities compared to multiport equivalents over a variety of cases. In all cases in over 4 years since our program’s initiation of SIPES usage we did not observe any incisional hernias.” There is not a reference, so, what study are they talking about? Furthermore, in the section “Myth 3: SIPES practice is only applicable to simple cases” the discussion consists in the presentation of 3 surgeries performed by the authors, underlining the total confusion of the structure of this work.
- If the authors’ intent was to produce a real review of literature, the entire manuscript should be better structured since it totally lacks of Materials and Methods, Results and Discussion sections. The authors do not describe at all the methodology used to conduct the review, the criteria for articles’ selection, nor the results. In this way, the conclusions seems to be rash and not supported by evidence.
- Moreover, the review of literature should be conducted in accordance with PRISMA guidelines.
- There is also an issue about the numbered list of the manuscript’s structure: it starts with “1. Introduction” and that’s all; numbers 2, 3, … do not exist.
- Authors should underline the limitations and drawbacks of the manuscript

Author Response

We appreciate the time and efforts by the editor and reviewers made towards this manuscript. We have addressed all issues indicated in the review report, as outlined below, and believe the revised version meets the journal publication standards.

Response to Comments from Reviewer 2

Major points:

“The manuscript deals with an interesting argument, the single incision pediatric endoscopic surgery (SIPES) and its applications today, aiming to review biases against single incision surgery and discuss the evidence about them. In particular the authors discuss five common myths about SIPES techniques: restrictions to specialized training, increased morbidity outcomes, limitations against complex cases, increased operative lengths, and increased operative costs. This kind of structure could be innovative and catchy, however, reading the manuscript, some major concerns have emerged:

It is not clear at all if the authors intended to write a review of literature, a case report or an article presenting their case series, resulting in a totally confusing work. Indeed, the authors presented the manuscript as a review of literature but, for example in rows 112-115 they write “In our own experience at a quarternary institution with surgical trainees involved in over 270 cases, we have not shown any increases in conversions to open (1.10%), SSI (3.3%) or otherwise morbidities compared to multiport equivalents over a variety of cases. In all cases in over 4 years since our program’s initiation of SIPES usage we did not observe any incisional hernias.” There is not a reference, so, what study are they talking about? Furthermore, in the section “Myth 3: SIPES practice is only applicable to simple cases” the discussion consists in the presentation of 3 surgeries performed by the authors, underlining the total confusion of the structure of this work. If the authors’ intent was to produce a real review of literature, the entire manuscript should be better structured since it totally lacks of Materials and Methods, Results and Discussion sections. The authors do not describe at all the methodology used to conduct the review, the criteria for articles’ selection, nor the results. In this way, the conclusions seems to be rash and not supported by evidence. Moreover the review of literature should be conducted in accordance with PRISMA guidelines. There is also an issue about the numbered list of the manuscript’s structure: it starts with “1. Introduction” and that’s all; numbers 2, 3, … do not exist. Authors should underline the limitations and drawbacks of the manuscript.

Response:

Thank you for your efforts in careful review of our report along with the valuable input.

and 3: We apologize for any confusion of intent that may have been perceived in this manuscript. This manuscript was intended to be a more traditional narrative literature review, instead of the systematic review perceived. As the reviewer points out, this manuscript aims to review biases against single incision surgery and discuss the evidence about them. While a systematic review would have a summed up the full literature available on each narrow, specific clinical question touched upon, the multiple themes encompassed in this review do not lend themselves to that particular methodology’s use over each individual focus for this manuscript’s setting. In the setting of a systemic review or meta-analysis PRISMA guidelines would have been followed. Instead, we sought a study design that lent itself to address the broader topics covered, where many different study designs have been applied and used. Specifically, we sought to highlight the array of evidence of the biases/preconceptions regarding SIPES, with specific focus on the updates in literature over the evolution of the SIPES technique adaptation, as well as misapplications of literature or even the gaps in literature that exist. The number “1. Introduction” placed before the introduction was an edit not in the submitted manuscript and has been removed from this version. As suggested by the reviewer, limitations were added to the concluding paragraph (lines 298-302), as well as highlighted throughout the manuscript as appropriate (lines: 115-119, 124-129, 130-134, 227-228, 254, 260-261, 263, 265-266, 282-284, 287-289).

Round 2

Reviewer 1 Report

Thanks for submitting improved manuscript. The general status of revised manuscript was improved. 

Reviewer 2 Report

The authors just limited to explain their intent in writing this work and only made some minor (and substantially irrelevant) modifications to the manuscript.

Therefore, I remain convinced that this "narrative review", structured in this way, does not add anything scientifically relevant to the current literature.
